# Horizontal Alignment Security Design Theory and Application of Superhighways

**Yu-Long Pei** [1,2,*], **Yong-Ming He** [1,3,*], **Bin Ran** [3,4], **Jia Kang** [1,2] and **Yu-Ting Song** [1,2]

1   School of Transportation, Northeast Forestry University, Harbin 150040, China; kangjia@nefu.edu.cn (J.K.);
    songyuting@nefu.edu.cn (Y.-T.S.)
2   Transport Research Centre, Northeast Forestry University, Harbin 150040, China
3   Department of Civil and Environmental Engineering, University of Wisconsin-Madison, Madison, WI 57305,
    USA; yhe275@wisc.edu
4   Southeast University-University of Wisconsin Intelligent Network Transportation Joint Research Institute,
    2312 Engineering Hall, 1415 Engineering Drive, Madison, WI 53706, USA
*   Correspondence: peiyulong@nefu.edu.cn (Y.-L.P.); hymjob@nefu.edu.cn (Y.-M.H.);
    Tel.: +86-136-33602189 (Y.-M.H.)

**Abstract:** In China, the maximum design speed of highways is 120 km/h, which first appeared in the *Highway Engineering Technical Standard (Trial)* in 1951. However, vehicle performance, road design, and construction technology have been greatly improved over the past 68 years. To adapt to the development demands of highway design speeds above 120 km/h in the future, it is urgent to study superhighway alignment design theory. Therefore, the horizontal alignment security design theory of superhighways was developed in this paper. First, the definition, classification, and construction mode of a superhighway and suitable vehicles of different grades are presented. Then, the lengths of straight lines were limited to reduce driving fatigue. Next, the minimum radii of circular curves were calculated based on driver characteristics and stress analysis of operating vehicles. Finally, the minimum lengths of transition curves were calculated based on the centrifugal acceleration of the operating vehicles, the travel time, and the passenger visual characteristics. The calculation and analysis results show that the superhighway linear features conform to the vehicle operating characteristics, and can ensure the safety of driving.

**Keywords:** superhighway; security design; horizontal alignment; circular curve; design speed

## 1. Introduction

Superhighways are highways with a design speed higher than 120 km/h. Superhighways are different from ordinary highways. To ensure the operation safety of superhighways, the pavements are flatter, the routes are smoother, and the facilities are more complete.

In China, the maximum design speed of highways is 120 km/h, which first appeared in the Highway Engineering Technical Standard (Trial) in 1951 [1]. Over the past 60 plus years, vehicle performance [2], road design, and construction technology have been greatly improved [3]. The maximum design speed no longer meets the needs of reality, and developed countries have been increasing highway speed limits [4]. For example, there are maximum speed limits of 137 km/h (85 mile/h) in parts of the United States [5], 130 km/h in France, Switzerland, and Austria, and 150 km/h in Italy, and some highways in Germany do not even have speed limits [6]. However, research related to highway design speeds higher than 120 km/h is still mostly lacking in China [7], and there are very few studies from foreign countries [8].

By the end of 2018, the total highway mileage in China had exceeded 140,000 km, ranking first in the world for nine years, and we have also built a number of super projects. For example, the ten longest

cross-sea bridges in the world, five of them belong to China, and the Hong Kong–ZhuHai–Macao Bridge ranked first. In 2018, automobile output in China reached 28.08 million, accounting for 32.52% of the world's 86.34 million. With the progress of automotive technology and the development of road construction technology, it is possible to improve the design speed of highways and to construct higher grade highways in China. To ensure the sustainable development of and adapt to the development demands of highway design speeds above 120 km/h in the future, it is of important practical significance to study the horizontal alignment design theory of superhighways [9].

In 2016, we proposed the concept of superhighways for the first time [10]. Two years later, in 2018, the first superhighway began to be built in China. The first superhighway being built in China runs from Hangzhou to Ningbo and will come into service before the 2022 Asian Games in Hangzhou. This indicates that the "superhighway" has entered the engineering practice. In 2019, the superhighway from Beijing to Xiongan with autonomous driving lanes was approved. The development of superhighways has far exceeded expectations. Therefore, there is an urgent need for research related to superhighways. To address this gap, we studied the feasibility, safety, and economy of superhighways in studies as follows.

In 2016, the feasibility and necessity of superhighways were demonstrated. First, the concept of superhighways was proposed, and the technical grade of superhighways was divided. Then, the feasibility of the superhighway was demonstrated. Finally, the necessity of superhighway development was studied. This paper describes a broad blueprint [10]. In 2017, the feasibility of superhighways was demonstrated from the aspects of vehicle technical conditions and road technical conditions. In the same year, the safety support and economic evaluation of superhighways were studied. The capacity of superhighways was researched from the perspective of traffic operation characteristics. The research results showed that autotrain special superhighways based on autonomous driving technology can not only improve the design speed of superhighways, but also improve the capacity of superhighways [11]. In 2018, an economic evaluation of superhighways based on travel costs was published. On the basis of analyzing the cost of ordinary expressways and high-speed railways, this paper estimated the cost of superhighways. Based on this, the toll standards of superhighways at all levels were estimated by referring to the construction costs and toll collection standards of ordinary expressways. The results show that superhighway travel has certain economic advantages. In 2018, a paper about the horizontal alignment design theory of superhighways was published. Based on the analysis of driving forces, the minimum radius of circular curves was determined, and the minimum length of easing curves was determined according to centrifugal acceleration, running time, and occupant visual characteristics. The results show that increasing the radius of horizontal curves, decreasing the slope of vertical curves and adjusting the design parameters of relaxation curves can meet the requirements of the safe operation of superhighways [12]. In 2019, an environmental and economic evaluation of superhighways based on travel costs was published. On the basis of fully analyzing the cost of various transportation modes, this paper estimated the construction and operation costs of superhighways. This study shows that travel by superhighways is cheaper than travel by air and cheaper than some high-speed rail fares [13].

In addition, in 2018, two Chinese scientists, Chen and Xu, proposed alternatives for the future development of superhighways by applying Strengths Weaknesses Opportunities Threats (SWOT) analysis to the characteristics, advantages and disadvantages, opportunities, and threats of the external environment during the construction and operation management of superhighways [14]. In 2019, Zhao Youchao, Mao Honggri, and Liu Jiangdong, researched the safety plane curve radius of superhighways. They proposed the safety requirements of superhighways, established an obstacle identification model, and calculated the safety plane curve radius based on research on expressway traffic accidents, which provided a basis for the research on superhighways and their plane curves [15].

Little research has been done on superhighways in the world, but many studies have been done on the plane linear design of ordinary highways. Hamilton and Himes [16] designed consistency in the context of highway and street design by referring to the conformance of highway geometry to

driver expectancy. Their study explored the relationships between alternative measures of horizontal alignment design consistency and the expected number of roadway departure crashes along horizontal curves on rural, two-lane, two-way roads. The authors analyzed 854 horizontal curves on rural two-lane highways in Indiana and Pennsylvania using data obtained from the SHRP 2 Roadway Information Database (RID) 2.0. Relationships between measures of design consistency and the expected number of roadway departure crashes were explored using a negative binomial regression modeling approach. The results indicated a relationship between the frequency of roadway departure crashes on a study curve and the radii of upstream and downstream curves. The ratio of the length of upstream and downstream tangents relative to a study curve radius was also statistically significant in Pennsylvania. Such findings are intuitive given the concept of design consistency and represent advancement of existing predictive methods in the AASHTO (American Association of State Highway and Transportation Officials) Highway Safety Manual, which estimate the expected number of crashes on a segment as a function of the characteristics of only that segment. Xu and Lin [17] proposed a new alignment design method that can take into account the typical handing patterns (driving styles) of human drivers and can pay special attention to dangerous driving behaviors. The core of the proposed method is forecasting the trajectories of typical direction control patterns within roadway width, which can be used by drivers. Then, the driving speed of typical speed control patterns is forecast on the basis of the curvature of the preview trajectory just determined. A mathematical programming method was used in this study, whereby objective functions and constraints were developed to model the typical driving patterns. This study provided five direction control patterns and four-speed control patterns to designers so that they can select an appropriate pattern to predict the trajectory and speed for the designed road. Ultimately, the trajectory and speed are used to control the geometric features of the road. Geometric features can determine the driveway shape, such as curve radius, deflection angle, spiral length, tangent length, and roadway/lane/shoulder width, and any or all of these can be adjusted by the designer. Additionally, as more than one driving pattern is possible, vehicle performance, driving stability, and ride comfort restrictions are introduced to trajectory/speed decision-making. This new method more closely approximates real-world driving than conventional methods. The application example shows that the proposed method is especially suitable for the horizontal alignment design of low/medium design speed highways that traverse rugged terrain. Bosurgi and Pellegrino [18] developed an optimization procedure using genetic algorithms (GAs) for selecting the different parameters of the PPC (Polynomial Parametric Curve). In particular, a specific algorithm defines the parameter values to minimize an appropriate fitness function. In addition, the final PPC can be examined from a dynamic point of view to evaluate compliance with comfort and safety conditions. Moreover, to simplify the geometric representation and the calculation of the dynamic variables of the PPC, these researchers used computer software and a specific and innovative routine to specifically develop the procedure.

The remainder of the paper is organized as follows. An overview of superhighways is presented in Section 2. In Section 3, the three elements of horizontal alignment are analyzed. In Sections 4–6, the lengths of straight lines, radii of circular curves, and lengths of the transition curves of superhighways are analyzed and calculated. Section 7 closes the paper with conclusions and further work.

## 2. Overview of Superhighways

### 2.1. The Background of Superhighways

In 2015, several professors from Tongji University and I went to Europe to attend an international conference on traffic safety, and found that the maximum speed limit in some countries is 130 km/h, and some highways have no speed limit in Germany. After returning to China, I began to investigate the origin of highways expressway speed limit, and found that it first appeared in the design code published in 1951 with surprise. After several expert demonstrations, it is believed that superhighways are feasible in China. Therefore, the research of superhighways is proposed.

The current highways in China use 120 km/h as the maximum design speed. Therefore, a superhighway is defined as a highway with a design speed higher than 120 km/h. A superhighway is also a highway and uses higher technical indicators than the *Highway Engineering Technical Standard* (JTG B01-2014).

### 2.2. Grade Division of Superhighway

According to the road use function, design speed, and definition of a superhighway, roads are divided into highways and common roads, and then the highways are divided into superhighways and common highways.

Among them, common highways and common roads correspond to highways and roads, respectively, from grade one to grade four in the *Highway Engineering Technical Standard* (JTG B01-2014), and the design speed has not changed. According to the design speed, superhighways are divided into three grades. Therefore, the new standard of road grade division is shown in Tables 1 and 2.

**Table 1.** New standard for road grade division.

| Road Grade | Highway | | | | Road | | | | | | | |
|---|---|---|---|---|---|---|---|---|---|---|---|---|
| | Superhighway | Common Highway | | | One | | Two | | Three | | Four | |
| Design speed (km/h) | >120 | 120 | 100 | 80 | 60 | 100 | 60 | 80 | 40 | 60 | 30 | 40 | 20 |

**Table 2.** New standard for superhighway grade division.

| Highway Grade | Superhighway | | | | | | | | | Common Highway | | |
|---|---|---|---|---|---|---|---|---|---|---|---|---|
| | Grade Three | | | Grade Two | | | Grade One | | | | | |
| Design speed (km/h) | 180 | 140 | 120 | 160 | 140 | 120 | 140 | 120 | 100 | 120 | 100 | 80 | 60 |

### 2.3. Comparison of Different Superhighway Levels

According to the classification above, the service objects, construction methods, and years of the implementation for different levels of superhighways are shown in Table 3.

**Table 3.** Comparison of different superhighway levels.

| Superhighway | Service Objects | Construction Methods | Years of Implementation |
|---|---|---|---|
| Grade one | For passenger vehicles and trucks | Reconstruction on existing highways | 15 years |
| Grade two | Only for passenger vehicles | Construction refers to passenger dedicated railway lines | 30 years |
| Grade three | Only for automatic driving vehicles | Construction refers to high speed railway lines | 40–50 years |

## 3. Superhighway Horizontal Alignments

### 3.1. Horizontal Alignment

The road is a ribbon of a three-dimensional entity, the space form of the road midline is called the route, and its projection line in the horizontal plane is known as the horizontal alignment of the road.

The route design is divided into plane line design, longitudinal section design, and cross-section design. Plane line design is the most important component of superhighway alignment design. The plane linear design theory contains three design elements.

### 3.2. Three Elements of Horizontal Alignment

There are three kinds of relationships between the steering wheel plane and the body longitudinal axis plane of a vehicle: the angle is 0, constant, or variable.

According to the above three kinds of relationships, the running track lines are as follows:

(1) The curvature is zero, and the radius of the running track lines is infinite, so the line is a straight line. As shown in Figure 1.

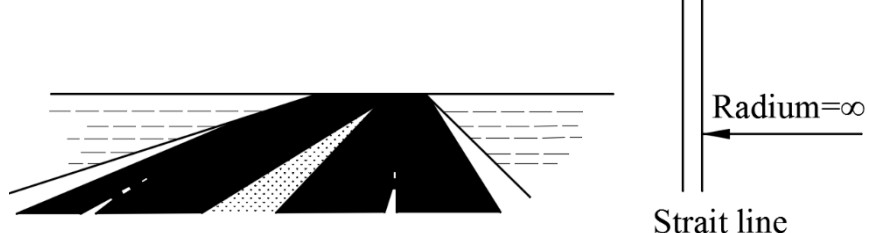

**Figure 1.** Straight line.

Straight lines are mainly used to connect circular curves and relief curves. For example, connect a circle curve to a circle curve, a circle curve to a transition curve.

(2) The curvature is a constant, and the radius of the running track lines is also a constant, so the line is a circular curve. As shown in Figure 2.

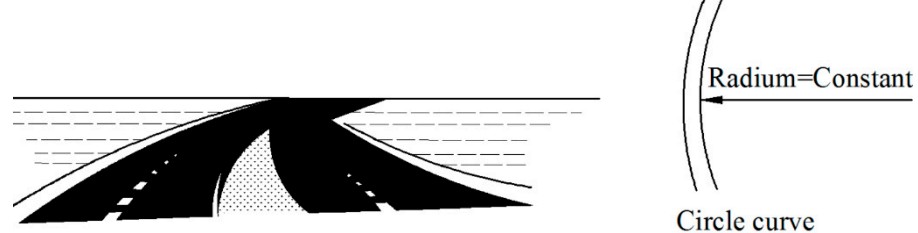

**Figure 2.** Circular curve.

A circle curve is mainly used for turning, and to change the direction of a vehicle.

(3) The curvature is variable, changing from zero to a constant, and the radius of the running track lines also varies, changing from infinity to a constant, so the line changes from a straight line to a circular curve. As shown in Figure 3.

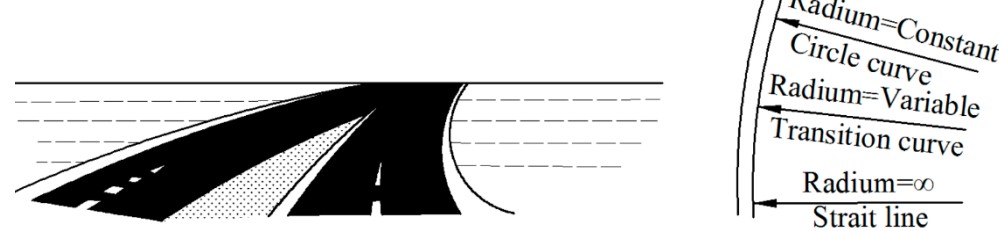

**Figure 3.** Straight line changing to circular curve.

A transition curve is used to connect a straight line to a circular curve, or a circular curve to a straight line, for gradually changing the direction of a vehicle.

The design elements of the plane line of the superhighway and common road are the same: straight lines, circular curves, and transition curves. Therefore, these components can also be called the three design elements of plane line design. To simplify the design, only two kinds of elements, straight lines, and circular curves, are used to design low-grade roads. There is no uniform regulation to restrict the proportion and frequency of use of plane line elements when designing common highways, as long as the use is reasonable, and each element if configured properly can meet the driving requirements. Superhighway plane line element design must be based on human visual and psychological factors, the condition of terrain and the road technical grade.

## 4. Length Restrictions on Straight Lines

### 4.1. The Maximum Length of a Straight Line

In the design of traditional highways, straight lines are widely used, but in the majority of cases, long straight lines are difficult to coordinate with the terrain. Improper length not only destroys the continuity of the overall alignment of the road but also cannot achieve coordination of the alignment design itself.

Long straight lines can easily make drivers feel bored and tired and cause difficulty in accurately visualizing the distance between two vehicles, generating driver impatience, the desire to pull out of the straight line as soon as possible and repetitive increases in speed, resulting in serious speeding, which could easily lead to the occurrence of traffic accidents. Therefore, when a straight line is used, it must not be too long.

According to foreign research results, for road design speed higher than or equal to 60 km/h, the maximum straight line length should be the distance traveled over approximately 70 s at a design speed that is equivalent to a length of 20 *V*. The vehicles traveling on superhighways of grade one and grade two are controlled by drivers and therefore need to consider the factors above. Superhighways of grade three are special roads for self-driving cars, and the maximum length of straight lines on these roads is not restricted. The maximum straight-line lengths of superhighways are shown in Table 4.

**Table 4.** Length limit for straight lines.

| V (km/h) | | Grade Three | | | Grade Two | | | Grade One | | |
|---|---|---|---|---|---|---|---|---|---|---|
| | | 180 | 160 | 140 | 160 | 140 | 120 | 140 | 120 | 100 |
| Maximum (m) | | - | - | - | 3200 | 2800 | 2400 | 2800 | 2400 | 2000 |
| Minimum (m) | Same | 1080 | 960 | 840 | 960 | 840 | 720 | 840 | 720 | 600 |
| | Reverse | 360 | 320 | 280 | 320 | 280 | 240 | 280 | 240 | 200 |

### 4.2. The Minimum Length of a Straight Line

(1) The minimum length of straight lines between two curves that turn in the same direction inserting a short straight line between the two curves that turn to the same direction is not reasonable. Visually, it is easy to see the straight line and the two curves at the ends as an illusion of a reverse curve or even see the two sections of the curve as one when the straight line is too short. This design destroys the continuity of the overall alignment of the road and easily causes driver operation error. Therefore, this design should be avoided. Because the defect generated by this linear combination easily produces an illusion for the driver, it is necessary to restrict the minimum length of the straight line between curves that turn in the same direction. Then, the adjacent curves ahead will not be visible to the driver at the same time, avoiding the above shortcomings. In view of the reasons above, the *Highway Engineering Technical Standards* (JTG B01-2014) and *Specification for Highway Routes* (JTG D20-2006) recommend that the advisable minimum length (m) between two curves turning in the same direction is not shorter than 6 times the design speed (km/h).

The requirements of high-speed (60 km/h or higher) roads should be implemented as much as possible; the design speed of superhighways of grades one and two is far higher than 120 km/h and the driving on these roads is controlled by humans, so the requirements must be guaranteed. Superhighways of grade three are special highways for self-driving cars and are not restricted by the minimum length restrictions. The minimum length of the straight line between curves turning in the same direction is shown in Table 4.

(2) The minimum length of the straight line between two curves that turn in opposite directions

When the two curves turn in opposite directions with no transition curve, taking into account the need to implement a wider transition section with superelevation, as well as the driver's steering operation needs, a length of straight line should be designed as a widened or superelevated transition section. *The Highway engineering technical standards* (JTG B01-2014) and *The Specification for Highway*

*Routes* (JTG D20-2006) suggest that the minimum length (m) of the straight line between two curves that turn to in opposite directions should not be shorter than 2 times the driving speed (km/h). Superhighways of grade three are special highways for self-driving cars and are not restricted by the minimum length restriction. The minimum length of the straight line between curves turning in opposite directions is shown in Table 4.

## 5. Determining the Radius of a Circular Curve

### 5.1. Vehicle Horizontal Stress Analysis When Operating

Vehicles operating on a circular curve are affected by gravity and centrifugal force [19]. Due to the production of centrifugal force, two hazards may arise when vehicles are operating on a curve: slipping outwards and overturning [20]. However, because modern automobile manufacturing standards minimize the center of gravity of vehicles, the danger of overturning is very small, and overturning occurs after slip, so only slip needs to be restricted [21]. As shown in Figure 4.

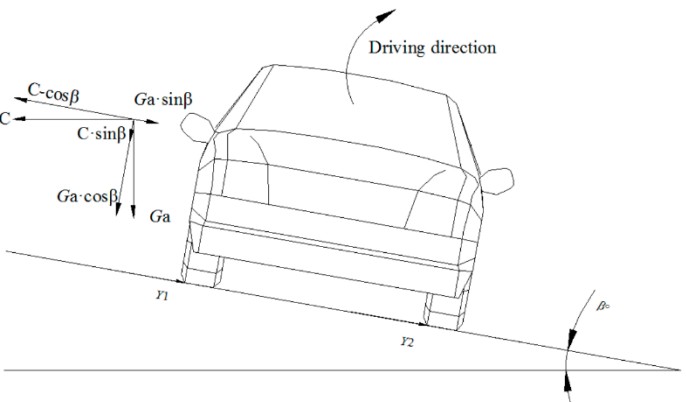

$\beta$—transverse slope angle of road; $h_g$—gravity center of vehicle; $B$—wheelspan; $C$—Centrifugal force; $Ga$—Vehicle gravity

**Figure 4.** Force analysis on a vehicle while driving on transverse slope.

Figure 4 shows that the lateral force generated by a vehicle operating on a horizontal curve $Y$ is given by Equations (1) [22] and (2) [23].

$$Y = C \times \cos\beta \pm G_a \times \sin\beta \tag{1}$$

Usually, the angle $\beta$ is small, and then $\cos\beta \approx 1$, and $\sin\beta \approx \tan\beta = i_0$

$$Y = C \pm G_a \times i_0 \tag{2}$$

where $i_0$ is the transverse slope of the road surface, "$\pm$" is "+" or "−", "+" indicates that the component of the centrifugal force parallel to the pavement and the force of gravity on the vehicle act in the same direction [24], that is, the vehicle operates on the outside of the road, and the road does not have a double slope with super elevation; "−" indicates the opposite of the above, that is, the vehicle runs on the inside of the road, and the road does not have a double slope with super elevation.

The centrifugal force C is calculated by Equation (3):

$$C = \frac{G_a}{g} \times \frac{v^2}{R} \tag{3}$$

where $v$ is the vehicle speed, m/s; $g$ is the acceleration due to gravity, m/s$^2$; and $R$ is the road curve radius, m.

Substitute Equation (3) into Equation (2), which yields Equation (4).

$$Y = \frac{G_a \times v^2}{g \times R} \pm G_a \times i_0 = G_a \times \left( \frac{v^2}{g \times R} \pm i_0 \right) \tag{4}$$

The ratio of the lateral force $Y$ to vehicle weight Ga is called the lateral force coefficient $\mu$, that is Equation (5).

$$\mu = \frac{Y}{G_a} = \frac{v^2}{g \times R} \pm i_0 \tag{5}$$

The vehicle lateral stability does not depend on the absolute value of $Y$, which depends on the relative lateral force per unit mass of the vehicle.

$R$ can be obtained from the above equation, as given in Equation (6).

$$R = \frac{v^2}{g \times (\mu \pm i_0)} \tag{6}$$

Through stress analysis, deduce that the curve radius value to ensure no slippage is given by Equation (7).

$$R = \frac{V^2}{127 \times (\mu \pm i)} \tag{7}$$

where $V$ is the vehicle speed (km/h), $\mu$ is the lateral force coefficient, and $i$ is the road transverse slope. When a vehicle operates inside of the curve, we define Equation (6) as "+", and when a vehicle operates outside of the curve, we define Equation (6) as "−".

When the grade of the superhighway is decided, the design speed is a fixed value, and then the curve radius $R$ is only related to the lateral force coefficient $\mu$ and the transverse slope $i$.

### 5.2. General Minimum Radius of the Circular Curve

Referring to domestic and international experience, we substituted the $\mu$ and $i$ from Table 4 into Equation (6) and obtained results in rounded up integer multiples of 50, and the general minimum radius ($R_{\min}$) of the superhighway is shown in Table 5.

**Table 5.** General minimum radii of circular curves.

| V (km/h) | Grade Three | | | Grade Two | | | Grade One | | |
|---|---|---|---|---|---|---|---|---|---|
| | 180 | 160 | 140 | 160 | 140 | 120 | 140 | 120 | 100 |
| u (%) | 4 | 4 | 5 | 4 | 5 | 5 | 5 | 5 | 5 |
| i (%) | 5 | 5 | 6 | 5 | 6 | 6 | 6 | 6 | 6 |
| $R_{\min}$ (m) | 2350 | 1850 | 1450 | 1850 | 1450 | 1050 | 1450 | 1000 | 700 |

Note: The results are given in integer multiples of 50.

The above calculation results consider the passenger comfort and safety when a vehicle operates on the curve at the design speed or close to the speed, and the engineering quantities are not increased too much under conditions where the terrain is complicated.

### 5.3. Circular Curve Limited Minimum Radius

According to Equation (7), using the maximum lateral force coefficient $\mu$ and the road cross slope $i$ provides the radius ensures a vehicle will not sideslip, that is, Equation (8).

$$R_{\min} = \frac{V^2}{127 \times (\mu_{\max} \pm i_{\max})} \tag{8}$$

where $V$ is the design speed (km/h), $\mu_{max}$ is the maximum allowable lateral force coefficient, and $i_{max}$ is the largest superelevation rate.

The maximum lateral force coefficient and superelevation according to the existing design technical standards, which consider passenger comfort and safety, are shown in Table 5. The results calculated for these values using Equation (6) are shown in Table 6.

**Table 6.** Limited minimum radii of circular curves.

| V (km/h) | | Grade Three | | | Grade Two | | | Grade One | | |
|---|---|---|---|---|---|---|---|---|---|---|
| | | 180 | 160 | 140 | 160 | 140 | 120 | 140 | 120 | 100 |
| | u (%) | 8 | 8 | 10 | 8 | 10 | 10 | 10 | 10 | 10 |
| | i = 0.04 | 2050 | 1600 | 1100 | 1600 | 1100 | 800 | 1100 | 800 | 500 |
| R (m) | i = 0.05 | 1950 | 1550 | 1050 | 1550 | 1050 | 800 | 1050 | 800 | 500 |
| | i = 0.06 | 1850 | 1450 | 1000 | 1450 | 1000 | 750 | 1000 | 750 | 450 |

Note: The results are given in integer multiples of 50.

*5.4. Minimum Radius with No Superelevation*

Superelevation may not be implemented for curve sections where the circular curve radius is larger than a certain value, allowing the use of a crown slope as in the straight sections. Considering the driving comfort, the lateral force coefficient should be controlled to the minimum. According to *The Specifications for the Design of Highway Routes* (JTG D20-2006), the lateral force coefficient value ranges from 0.035 to 0.050.

Because there is no superelevation, the road cross-section is a crown slope. According to the stress analysis, a vehicle operating on the outside of the curve is in the worst case. Then, the crown slope $i = i_0$, so according to Equation (6), the minimum radius can be obtained by Equation (9).

$$R_{\min no} = \frac{V^2}{127 \times (\mu \pm i)} \tag{9}$$

where $R_{minno}$ is the minimum radius with no superelevation in units of m and i0 is the crown slope with values ranging from 0.015 to 0.025.

According to Equation (8), the calculated minimum radii are shown in Table 7.

**Table 7.** Minimum radii with no superelevation.

| V (km/h) | | Grade Three | | | Grade Two | | | Grade One | | |
|---|---|---|---|---|---|---|---|---|---|---|
| | | 180 | 160 | 140 | 160 | 140 | 120 | 140 | 120 | 100 |
| | u (%) | 4.5 | 4.5 | 5.0 | 4.5 | 5.0 | 5.0 | 5.0 | 5.0 | 5.0 |
| | i₀ = 1.5 (%) | 4650 | 3150 | 2400 | 3150 | 2400 | 1750 | 2400 | 1750 | 1150 |
| R_min (m) | i₀ = 2.0 (%) | 4300 | 2900 | 2250 | 2900 | 2250 | 1650 | 2250 | 1650 | 1050 |
| | i₀ = 2.5 (%) | 3950 | 2700 | 2100 | 2700 | 2100 | 1550 | 2100 | 1550 | 1000 |

Note: The results are given in integer multiples of 50.

## 6. Determining the Length of the Transition Curve

The curvature is constantly changing as vehicles transition from linear to circular curves, and the section with changing curvature is called the transition curve. Clothoids, cubic parabolas, and twisted pair lines are commonly used for transition curves. In recent years, quintic curves and driving aesthetics curves have also been widely used. Considering convenience and habits of use, *Highway Engineering Technical Standards* (JTG B01-2014) uses cyclotron lines as transition curves.

Transition curves must be long enough to avoid centrifugal acceleration growth or the driver having to turn the steering wheel too fast and should make driving safe, comfortable, and linear optimum smooth. The minimum length of a transition curve, in general, should meet the centrifugal acceleration, travel time, and visual condition requirements.

### 6.1. The Change Rate of Centrifugal Acceleration should not be too Fast

When a vehicle operates on a transition curve, the centrifugal acceleration generated by centrifugal force is $a = v^2/\rho$, and the time from the beginning to the end of the transition curve is $t$ (s).

During this process, the radius of curvature $\rho$ uniformly changes from 0 to $R$, and the centrifugal acceleration uniformly increases from 0 to $v^2/R$; then, the growth rate of acceleration generated is Equation (10).

$$a_s = \frac{a}{t} = \frac{v^2}{R \times t} \tag{10}$$

Assuming that the vehicle driving at a constant speed, then $t = l_s/v$, and $a_s = v^3/Rl_s$, which yields Equation (11).

$$l_s = \frac{v^3}{R \times a_s} \tag{11}$$

We convert $v$ (m/s) to $V$ (km/h) and obtain Equation (12).

$$l_s = 0.0214 \times \frac{V^3}{R \times a_s} \tag{12}$$

where $V$ is the design speed (km/h), $a_s$ is the average rate of change of the centrifugal acceleration (m/s$^3$) and $R$ is the radius of the circular curve (m).

$a_s$ is an indicator to test the transition degree of the transition curve, generally this parameter is called the "transition factor", the units are m/s$^3$, and the passengers' comfort level depends on this factor.

The passengers' comfort level depends on $a_s$. As early as 1909, a British person named Shortt suggested that $a_s$ should not be greater than 0.6 m/s$^3$ so that the passengers will not feel that the vehicle is turning. Actually, the lateral force sensed is not all centrifugal force; this value does not include the centrifugal force offset by superelevations, so its interpretation is incomplete, but the relationship is still available.

The length of the transition curve can be obtained for a certain vehicle speed and certain curve radius when $a_s$ is chosen. The value selection of $a_s$ should mainly be based on the requirements of driving, including being able to manipulate the vehicle leisurely and keep the vehicle in its lane more accurately. If the transition curve $L_s$ is too short and the speed is too high, the driver will be tense and frantic while turning the steering wheel rapidly, even causing the vehicle to leave the line and cause traffic accidents. Practice has proven that as long as the value of $a_s$ can meet the requirements of driving, it can meet the comfort requirements of passengers.

For the value of $a_s$, there is no unified standard. The value Shortt proposed is 0.6 m/s$^3$, which is too large for high-speed roads and too small for low-speed roads. Generally, for high-speed roads, $a_s$ is 0.3 m/s$^3$ in the United Kingdom and 0.6 m/s$^3$ in the United States. For curves at an intersection in the United States, $a_s$ is 0.75 m/s$^3$ when the speed is 80 km/h, $a_s$ is 1.2 m/s3 when the speed is 32 km/h, and so on. In other words, for traditional highways with design speeds from 30 km/h to 120 km/h, the $a_s$ value ranges from 1.0 m/s$^3$ to 0.3 m/s$^3$. The design speed of superhighways is higher than 120 km/h, so these roads require the minimum limit of 0.3 m/s$^3$.

### 6.2. The Driving Time Should Not Be Too Short

Regardless of the parameters of the transition curve, drivers should be given enough time to adjust their direction. If the road turns too sharply, this can also make the passengers feel uncomfortable.

Therefore, the shortest travel time on transition curves should be limited. The shortest travel time on transition curves in China is 3 s, and thus, the minimum length of transition curves is given by Equation (13).

$$l_s = v \times t = \frac{V}{3.6} \times t = \frac{V}{1.2} \tag{13}$$

where $l_s$ is the minimum length of the transition curve (m), $v$ is the driving speed, t is the driving time (s), and $V$ is the design speed (km/h).

## 6.3. Meeting the Requirements of Visual Conditions

Research results and practice show that the minimum turning angle of transition curves is $\beta_1 = 3°10'59'' = 0.0556$ rad and the maximum turning angle $\beta_2 = 28°38'52'' = 0.5$ rad.

As $\beta = \frac{l}{2R} = \frac{A^2}{2R^2}$, $A^2 = R \times l$, then $A^2 = 2 \times R^2 \times \beta$, so $A = R\sqrt{2\beta}$

When $\beta = \beta_1$, $A = R\sqrt{2 \times 0.0556} = \frac{R}{3}$, and $l = \frac{A^2}{R} = \frac{R}{9}$

When $\beta = \beta_2$, $A = R\sqrt{2 \times 0.5} = R$, and $l = R$

Therefore, to make transitions linear, smooth, and coordinated, Equation (14) should be met.

$$l_s = \frac{R}{9} \sim R \tag{14}$$

*Highway Engineering Technical Standards* (JTG B01-2014) require the travel time on a transition section to be 3 s and limit the rate of change of centrifugal acceleration to 0.5–0.6 m/s$^3$. According to the highway design speed and corresponding grade, the minimum length transition curve can be calculated. The minimum length of the transition curve is shown in Table 8.

**Table 8.** Minimum lengths of transition curves when circular curve has a general minimum radius (Units: m).

| V (km/h) | | Grade Three | | | Grade Two | | | Grade One | |
|---|---|---|---|---|---|---|---|---|---|
| | 180 | 160 | 140 | 160 | 140 | 120 | 140 | 120 | 100 |
| $a_s$ restriction | 177 | 158 | 135 | 158 | 135 | 117 | 135 | 117 | 103 |
| Travel time restriction | 150 | 133 | 117 | 133 | 117 | 100 | 117 | 100 | 84 |
| Visual restriction   R/9 | 261 | 206 | 161 | 206 | 161 | 117 | 161 | 117 | 94 |
| Visual restriction   R | 2350 | 1850 | 1450 | 1850 | 1450 | 1050 | 1450 | 1050 | 850 |
| Minimum length | 265 | 210 | 165 | 210 | 165 | 120 | 165 | 120 | 95 |

Note: The results are given in integer multiples of 50.

The minimum length of the transition curve shown in Table 7 is only for a circular curve that has the general minimum radius. The situation is more complicated when the circular curve has the limited minimum radius or the minimum radius with no superelevation.

The limited minimum radius of a circular curve of the superhighway varies with superelevation $i$, and superelevation $i$ may be 0.06, 0.08, or 0.10. We take superelevation $i=0.08$ to calculate the limited minimum radius and then calculate the minimum length of the transition curve according to the method above; the results are shown in Table 9.

**Table 9.** Minimum lengths of transition curves when circular curve has a limited minimum radius (Units: m).

| V (km/h) | | Grade Three | | | Grade Two | | | Grade One | |
|---|---|---|---|---|---|---|---|---|---|
| | 180 | 160 | 140 | 160 | 140 | 120 | 140 | 120 | 100 |
| $a_s$ restriction | 260 | 225 | 217 | 225 | 217 | 190 | 217 | 190 | 164 |
| Travel time restriction | 150 | 133 | 117 | 133 | 117 | 100 | 117 | 100 | 86 |
| Visual restriction   R/9 | 178 | 144 | 100 | 144 | 100 | 72 | 100 | 72 | 50 |
| Visual restriction   R | 1600 | 1300 | 900 | 1300 | 900 | 650 | 900 | 650 | 450 |
| Minimum length | 260 | 225 | 220 | 225 | 220 | 190 | 220 | 190 | 165 |

Note: 1. The results are given in integer multiples of 50; 2. The limited minimum radius is calculated for superelevation $i$ of 0.08.

The minimum radius of a circular curve without superhighway superelevation varies with crown slope $i$, and crown slope $i$ may be 0.015, 0.020, or 0.025. We take the crown slope $i = 0.020$ to calculate

the limited minimum radius and then calculate the minimum length of the transition curve according to the method above; the results are shown in Table 10.

**Table 10.** Minimum lengths of transition curves when circular curve is not set ultrahigh (Units: m).

| V (km/h) | Grade Three | | | Grade Two | | | Grade One | | |
|---|---|---|---|---|---|---|---|---|---|
| | 180 | 160 | 140 | 160 | 140 | 120 | 140 | 120 | 100 |
| $a_s$ restriction | 97 | 92 | 87 | 92 | 87 | 75 | 87 | 75 | 68 |
| Travel time restriction | 150 | 133 | 117 | 133 | 117 | 100 | 117 | 100 | 83 |
| Visual restriction　R/9 | 478 | 322 | 250 | 322 | 250 | 183 | 250 | 183 | 117 |
| R | 4300 | 2900 | 2250 | 2900 | 2250 | 1650 | 2250 | 1650 | 1050 |
| Minimum length | 475 | 325 | 250 | 325 | 250 | 185 | 250 | 185 | 120 |

Note: 1. The results are given in integer multiples of 50; 2. The limited minimum radius is calculated for crown slope $i$ of 0.020.

The calculated value is the minimum length of the easement curve, but in most cases, to obtain a comfortable and beautiful line shape and good visual effect, a longer transition curve is often used.

It should be pointed out that the minimum length of a transition curve calculated based on the change rate of centrifugal acceleration and superelevation decreases gradually with increasing curve radius. However, visually, it is hoped that the transition curve should increase correspondingly with the increment of the curve radius. Especially for superhighways, the curve should be adjusted to adapt to the terrain and landscape as much as possible to make the curve visually smooth.

## 7. Conclusions

First, this paper expounds upon the background of superhighways, introduces the concept and the grade of the superhighway, analyzes the road and vehicle performance adapted to superhighways of different grades, and proposes the construction method and the implementation period of each grade of superhighway. Then, the horizontal alignment is described, and the relevant factors that should be considered are the "Three Elements of Horizontal Alignment". Next, according to the driver's physiological and psychological characteristics, the length limit of the straight lines of superhighways of grade one and grade two is calculated. Superhighways of grade three are special roads for only automatic driving vehicles, and the maximum length and the minimum length of the straight lines are not restricted on these roads. The general minimum radius, the limited minimum radius, and the minimum radius without superhighway superelevation are deduced and calculated based on stress analysis while a vehicle is operating. Finally, according to the centrifugal acceleration, travel time, and visual characteristics of the passengers, the minimum length of the transition curves of superhighways is restricted.

Through the above research, we come to the following conclusions:

(1) When the designed speeds of superhighways are 140 and 160 km/h, the maximum lengths of straight lines are 2800 and 3200 m, respectively. There is no limit to the maximum straight-line length of a level three superhighway. When the designed speeds of superhighways are 140, 160, and 180 km/h, the shortest straight line lengths between the same-direction curves are 840, 960, and 1080 m, respectively. The minimum lengths between the opposite-direction curves are 280, 320, and 360 m.

(2) When the design speeds of superhighways are 140, 160, and 180 km/h, the general minimum curve radii are 1450, 1850, and 2350 m, respectively. The minimum limited radii are 1000, 1450, and 1850 m, respectively. The minimum radii with no superelevation are 2100, 2700, and 3950 m, respectively.

(3) When the designed speeds of superhighways are 140, 160, and 180 km/h, the minimum lengths of transition curves are 165, 210, and 265 m, respectively. The minimum lengths of transition curves when circular curve has a limited minimum radius are 220, 225, and 260 m, respectively. The minimum lengths of transition curves when circular curve is not set ultrahigh are 250, 325, and 475 m, respectively.

There is little research on designing superhighways with speeds exceeding 120 km/h in China. This paper only studied the plane alignment design parameters of superhighways, and the research depth

and breadth are far from sufficient. Our research contributed to construction of the first superhighway (from Hanghzou to Ningbo via Shaoxing), and contributed to the approval of the plan for the second superhighway (from Beijing to Xiongan). In the future, we will continue to study the longitudinal and transverse alignment of superhighways, after which we will devote ourselves to the study of the fuel consumption of superhighways.

**Author Contributions:** Conceptualization, Y.-M.H., Y.-L.P., and B.R.; data curation, J.K. and Y.-T.S.; formal analysis, Y.-M.H., K.J., and Y.-T.S.; funding acquisition, Y.-M.H. and Y.-L.P.; investigation, Y.-M.H., K.J., and Y.-T.S.; methodology, B.R. and Y.-L.P.; project administration, Y.-M.H. and K.J.; supervision, B.R. and Y.-L.P.; validation, Y.-M.H., B.R., and Y.-L.P.; writing—original draft, Y.-M.H. and K.J.; writing—review and editing, B.R. and Y.-L.P. All authors have read and agreed to the published version of the manuscript.

**Funding:** This research was funded by the Natural Science Foundation of China, grant number 71771047 and it also funded by the Natural Science Foundation of Heilongjiang Province, China, grant number LH2019E004.

**Acknowledgments:** We thank Shaoyan Li, laboratory director of Transportation College of Northeast Forestry University for providing help for the experiment. We thank Yang Cheng of the College of Engineering, University of Wisconsin-Madison for his suggestions regarding the experimental scheme.

**Conflicts of Interest:** None of the authors have any conflicts of interest. All the authors identify and declare that they have no personal circumstances or interests that may be perceived as inappropriately influencing the representation or interpretation of the reported research results. All the authors declare that none of the funders had any role in the design of the study; in the collection, analyses, or interpretation of data; in the writing of the manuscript, or in the decision to publish the results.

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
