# Peer review of "Horizontal Alignment Security Design Theory and Application of Superhighways"

_sustainability, doi:10.3390/su12062222_

Round 1

Reviewer 1 Report

This paper enters into a new topic “superhighways” or highways with a design speed over the current standards. The paper format, structure, and style are correct.

However, there are some points of critique:

1) The study seems to have applied the current equations of the road standards, but substituting the speed by a higher one. The rest of the assumptions remain as in the standards. However, little research is made about driving behavior at higher speeds, so the extrapolation maybe not correct. In particular, the assumption may be correct for curve radii (the model represents the dynamics of vehicles) but not for the distance between consecutive curves (the model reflects the behavior of drivers).

2) There is no discussion about the sustainability of superhighways. In fact, superhighways are simply a very high-quality highway where one can see, at least, the following consequences: they are more expensive than conventional roads, they consume more land, the motivate higher travel speeds and higher energy consumptions. The authors did not measure these negative points at all and wrote only about travel time savings or safety (which are also important).

3) An additional point could be to add automated vehicles into the discussion. The automated vehicles may (or may not) reduce perception and reaction time, and then reduce the sight distance requirements. Moreover, automated driving can alleviate the requirements about design consistency (e.g. by allowing any sequence of curve radii, since the expectation of drivers is not important anymore). With these effects, the roads for automated vehicles could be actually “smaller” in terms of land consumption and cost. This point was not addressed in the paper.

Other minor comments:

-Figures 2 and 3 are unclear (the image in perspective does not clarify anything).

-Line 224 “for self-driving cars are not restricted by…”, but Table 4 says that they are restricted.

Author Response

Response to Reviewer 1 Comments

Thank you very much for your comments and suggestions. We have tried our best to revise the manuscript. If there are still problems, please give us another chance to modify, we will try our best to modify, until you are satisfied.

Point 1: The study seems to have applied the current equations of the road standards, but substituting the speed by a higher one. The rest of the assumptions remain as in the standards. However, little research is made about driving behavior at higher speeds, so the extrapolation maybe not correct. In particular, the assumption may be correct for curve radii (the model represents the dynamics of vehicles) but not for the distance between consecutive curves (the model reflects the behavior of drivers).

Response 1: Thank you for your comments. This manuscript is the result of our first study on the horizontal alignment of superhighways, and it may have some flaws. We're running a lot of experiments to see if the assumptions make sense.

Point 2: There is no discussion about the sustainability of superhighways. In fact, superhighways are simply a very high-quality highway where one can see, at least, the following consequences: they are more expensive than conventional roads, they consume more land, the motivate higher travel speeds and higher energy consumptions. The authors did not measure these negative points at all and wrote only about travel time savings or safety (which are also important).

Response 2: Thank you very much for your suggestion. I added the discussion of sustainable development of superhighways in the second paragraph. The key point of this manuscript is to ensure the traffic safety of superhighways through the linear design of superhighways. Of course, in keeping with the study scope of the journal, adding discussion on。

Point 3: An additional point could be to add automated vehicles into the discussion. The automated vehicles may (or may not) reduce perception and reaction time, and then reduce the sight distance requirements. Moreover, automated driving can alleviate the requirements about design consistency (e.g. by allowing any sequence of curve radii, since the expectation of drivers is not important anymore). With these effects, the roads for automated vehicles could be actually “smaller” in terms of land consumption and cost. This point was not addressed in the paper.

Response 3: You can see the discussion on automated vehicles in line 225. Because there is no driver in automated vehicles while drive on superhighway of grade three, needn’t to consider visual fatigue, there is no maximum length limit on the strait line, and the results are list in table 4. But when we calculate the curve radius of superhighway, the stability of vehicles and the comfort of passengers are mainly taken into account, which has little to do with whether it is automatic driving, so automatic driving is not taken into account.

Point 4: Figures 2 and 3 are unclear (the image in perspective does not clarify anything).

Response 4: We redrew figures 2 and 3. In addition, we also added the use description of figures 1-3.

Point 5: Line 224 “for self-driving cars are not restricted by…”, but Table 4 says that they are restricted.

Response 5: I think you may have misunderstood us here. There is no maximum length limit on the strait line of superhighway of grade three, but the minimum length limits of the strait line still exist.

Reviewer 2 Report

On lines 426 through 428, the numbers for the general minimum radii should be 1450 m, 1850 m, and 2350 m. The minimum limited radii should be 800 m, 1150 m. and 2350 m. 

Author Response

Response to Reviewer 2 Comments

Thank you very much for your comments and suggestions. We have tried our best to revise the manuscript. If there are still problems, please give us another chance to modify, we will try our best to modify, until you are satisfied.

Point 1: On lines 426 through 428, the numbers for the general minimum radii should be 1450 m, 1850 m, and 2350 m. The minimum limited radii should be 800 m, 1150 m. and 2350 m. 

Response 1: I am very sorry, these are very stupid mistakes, I have corrected it.

Reviewer 3 Report

In the evaluated work, the authors presented the concept of increasing design speed on superhighways. Based on the analysis of available literature data and the results of external research presented in various reports, the authors conducted an analysis of meeting the set of formal requirements for vehicle traffic safety and driving comfort for drivers on horizontal curves and on long straight sections.

The important safety condition, which may be sliding the stopped vehicle off the road, on a curve with a cant and of = 10%, in winter conditions (ice, snow) has not been checked. This is an important criterion because winter conditions in China also occur.

The actual impact of high vehicle speeds, which are usually greater than the speed limits, on the consequences of road accidents is also not taken into account.

Author Response

Response to Reviewer 3 Comments

Thank you very much for your comments and suggestions. We have tried our best to revise the manuscript. If there are still problems, please give us another chance to modify, we will try our best to modify, until you are satisfied.

Point 1: In the evaluated work, the authors presented the concept of increasing design speed on superhighways. Based on the analysis of available literature data and the results of external research presented in various reports, the authors conducted an analysis of meeting the set of formal requirements for vehicle traffic safety and driving comfort for drivers on horizontal curves and on long straight sections

Response 1: Thank you very much for your comments. We will try our best to do better in the future research.

Point 2: The important safety condition, which may be sliding the stopped vehicle off the road, on a curve with a cant and of = 10%, in winter conditions (ice, snow) has not been checked. This is an important criterion because winter conditions in China also occur.

Response 2: Thanks for your suggestion, we adjusted the superelevation and recalculated.

Point 3: The actual impact of high vehicle speeds, which are usually greater than the speed limits, on the consequences of road accidents is also not taken into account.

Response 3: Yes, we haven’t taken into account on the consequences of road accidents caused by overspeed. But few people in China dare to speed because of the density of surveillance cameras on the highways. Once speeding will face serious penalties, so there are few speeding vehicles. In many countries the average speed on the highways is above the maximum speed limit, but in China is the opposite. For example, the average traffic speeds on U.S. highways are generally 3 - 8 % above the speed limit, in German they are generally 4-7% above the speed limit. But in China, highway traffic speeds are 4-9% below the maximum speed limit.

Reviewer 4 Report

The english writing and style needs to be improved.

P2, L 47: Use "was" instead of began to be.

P 2, L 52: Try to avoid I, we, ... in your sentences.

P2, L 61: " The research results showed...

P2, L79: Spell out the numbers, i.e.  two chinese scientists.

P2, L 79: Use "alternatives" instead of suggestions.

P2, L 83: There is no need to mention the affiliation of writers in the body of article.

P3, L 135: Move the definition of superhighways to introduction part before you start talking about these types of highways.

Figures 1,2, 3: Use higher quality pictures.

Table 4: Show the table in one page.

P 8, L 287: Try to avoid unnecessary repetitions (i.e. you have used the word "that" three times here)

P 12, L 432: This paper only studied...

Conclusions:  The discussion of results needs to be more  elaborated.

I think it would be beneficial to the readers if you include an example of the presented theory at the end of the article.

Author Response

Response to Reviewer 4 Comments

Thank you very much for your comments and suggestions. We have tried our best to revise the manuscript. If there are still problems, please give us another chance to modify, we will try our best to modify, until you are satisfied.

Point 1: The english writing and style needs to be improved.

Response 1: I have entrusted the polishing company to deal with it before submitting the paper. After I modified it this time, I had asked my American friend to help me modify.

Point 2: P2, L 47: Use "was" instead of began to be.

Response 2: The superhighway began to be built in 2018, but it has not been completed yet. Therefore, the original statement may be right.

Point 3:P 2, L 52: Try to avoid I, we, ... in your sentences.

Response 3: Thank you very much for your suggestion. I've revised, and I've revised all the similar statements in the manuscript.

Point 4: P2, L 61: " The research results showed...

Response 4: Thank you very much for your suggestion. I've revised.

Point 5: P2, L79: Spell out the numbers, i.e.  two chinese scientists.

Response 5: Thank you very much for your suggestion. I've revised.

Point 6: P2, L 79: Use "alternatives" instead of suggestions.

Response 6: Thank you very much for your suggestion. I've revised.

Point 7: P2, L 83: There is no need to mention the affiliation of writers in the body of article.

Response 7: Thank you very much for your suggestion. I've revised.

Point 8: P3, L 135: Move the definition of superhighways to introduction part before you start talking about these types of highways.

Response 8: Thank you very much for your suggestion. I've moved the definition of superhighways to introduction part.

Point 9: Figures 1,2, 3: Use higher quality pictures.

Response 9: Thank you very much for your suggestion. Figure 1- figure 3 has been redrawn.

Point 10: Table 4: Show the table in one page.

Response 10: Thank you very much for your suggestion. I have modified, and all the forms are on the same page, none of the forms take up two pages.

Point 11: P 10, L 287: Try to avoid unnecessary repetitions (i.e. you have used the word "that" three times here)

Response 11: Thank you very much for your suggestion. I deleted 2 unnecessary "that".

Point 12: P 12, L 432: This paper only studied...

 Response 12: Thank you very much for your suggestion. I've changed “studies” to “studied”.

Point 13: Conclusions:  The discussion of results needs to be more  elaborated.

 Response 13: Thank you very much for your suggestion. More results of the study are added to the conclusions, and the main results are all included now.

Point 14: I think it would be beneficial to the readers if you include an example of the presented theory at the end of the article.

Response 14: At the end of the conclusion, two examples of the development of superhighways in China were added.

Round 2

Reviewer 1 Report

 No new comments.